# Implemental delay as a mediator of the relationship between depression, anxiety, stress and school burnout

Katarzyna Markiewicz *[ID][*], Bożydar L. J. Kaczmarek[*]

Institute of Psychology, WSEI University, Lublin, Poland

[*] These authors contributed equally to this work.
* katarzyna.markiewicz@wsei.pl

## Abstract

### Introduction

Research following the COVID-19 pandemic points to many problems related to adolescents' mental condition, their coping with the organization of daily life and the implementation of school duties, which can lead to school burnout. It includes absenteeism, decreased motivation and academic performance, and a cynical attitude toward peers and teachers. A significant way to protect adolescents' mental health may be by teaching them to complete their daily duties. Still, an obstacle is the observable post-pandemic, increasing tendency to procrastinate, which can exacerbate the effects of school burnout. The study aimed to establish the impact of depression, anxiety and stress on school burnout mediated by procrastination, understood as implemental procrastination.

### Method

Three questionnaires were used: Depression, Anxiety and Stress Scale, Implemental Delay Scale, and School burnout scale. The study comprised 344 students (57,6% females) from Polish secondary schools (high schools and technical schools), aged 14–20 years (M = 16.69; SD = 1.74). The analysis of relationships between variables studied was performed with the mediation model.

### Results

Analyses confirmed the mediating effect of implemental delay on the relationship between depression, stress and school burnout. Unexpectedly, it turned out that the burnout resulting from parental pressure significantly weakened the value of the mediator (implemental delay). In contrast, the value of the predictors (depression and stress) increased. Anxiety was found to be a nonsignificant predictor of school burnout.

### Conclusion

The data show that stress, depression and procrastination influence school burnout, which depends mainly on the dimension of burnout. Awareness of these relationships can support

**Funding:** The author(s) received no specific funding for this work.

**Competing interests:** The authors have declared that no competing interests exist.

the creation of specialized prevention programs, allowing students to overcome their problems. At the same time, parents and teachers may help them understand the nature of observed disorders, which has little to do with laziness often attributed to young people.

## Introduction

Assessments of the mental state of adolescents conducted over the past few years show a growing problem. The paper published in *Jama Pediatrics* [1] shows that the percentage of children and adolescents experiencing anxiety and depression increased significantly between 2016 and 2020. In addition, there was a decline in overall health-related quality of life (HRQoL) scores during the COVID-19 pandemic, which did not subside after the lockdowns ended. A report of the Kaiser Family Foundation [2] revealed that more than 25% of high school students reported "deterioration in emotional health and cognitive well-being." It is critical due to the lack of proper medical and psychological help and daily care in educational institutions [3]. Orban, et al. [4] argue that protecting the mental health of adolescents can accomplished by the implementation of daily routine tasks. Students who learned to complete daily tasks are more likely to use active and positive coping strategies, which are effective in helping to control negative thoughts and/or reduce negative mental states [1,5,6] and consequently prevent the effects of premature burnout, referred to as school or educational burnout. The aim of the present study was to analyze the mediating role of implemental delay on the impact of depression, anxiety and stress on school burnout.

### Depression, anxiety, and stress

Depression has become a common, debilitating mental illness with mood disorders that threatens people's physical health. Uncontrolled and untreated, it causes a more significant decline in health than other chronic diseases [7]. Stress and anxiety, as experienced by students, have been defined from the perspective of school and academic requirements as threatening students' well-being [8]. Research indicates positive links between depression, anxiety, stress [9] and procrastination [10]. Depression, anxiety and stress, especially during adolescence, can significantly inhibit educational success, e.g., impair academic success [11,12], and lead to resignation resulting from inability to meet obligations [13]. Experiencing anxiety and severe stress in an exam or assessment situation increases tension, leading to an increased sense of insecurity and physiological [14] and behavioral disorders [15,16]. Among the behavioral consequences of perceived stress and anxiety, the tendency to avoid or delay action is cited [17].

### Procrastination

Steel [18] authored the first in-depth description of the nature of procrastination, which he defined as the unnecessary and unwanted irrational delay of mostly aversive tasks. It can refer to the decision-making (intention to act), implementation (execution of the action) or discharge (accounting for/completing the action) phases of the task [19–22]. The procrastinator is generally aware of the negative consequences, and realizes the irrationality of his behavior [23], yet is unable to stop this reaction [24–28]. Most of the research on procrastination deals with procrastination in terms of starting tasks on time, while the core of the problem is what happens in the execution process [29,30]. Procrastination at the stage of goal pursuit means doing something other than what one should do, impulsively redirecting actions to other

situational alternatives [31]. The inability to carry out a given intention, known as the gap between intention and action [31,32], is associated with destructive thoughts, experiencing feelings of anxiety, failure to monitor progress toward goals, competing goals, bad habits, and low willpower [33].

Although many studies [9,18,34–36] point to deficits in self-regulation and self-control as the source of procrastination behavior, the mechanism appears more complex. Some reports suggest that an essential predictor of procrastination behavior is unconscious and out-of-control or paroxysmal anxiety [37,38]. However, it is not clear whether anxiety is a predictor of procrastination or, conversely, procrastination is a predictor of anxiety [10]. Most people report the tendency to procrastinate at different times in one's life, yet 75% of students rate this problem as severe, and almost half of them procrastinate consistently and problematically [6,39–44]. It can impair health, well-being, and academic performance, make it difficult to perform routine duties, and, consequently, lead to school burnout.

## School burnout

The term school burnout, linked to the theory of occupational burnout [45], was popularized by Maslach [46,47]. It is associated with increased absenteeism, low motivation, decreased commitment to learning and, consequently, decreased academic performance, perceived exhaustion, and cynical attitudes toward peers and teachers [48]. Excessive demands inconsistent with students' learning needs have been cited as causes of burnout [49]. Given the negative impact that each of the factors (depression, anxiety, stress, procrastination and school burnout) listed above can have on educational achievement and well-being, it becomes crucial to understand their combined effect. It will enable the design of an effective social support system understood as caring and willing to assist in solving problems plaguing adolescents.

Studies found that depression, anxiety, and stress lead to negative consequences [50,51], including increased propensity for procrastination [9,10,17,52]. During adolescence, the maturing brain undergoes structural and functional transformations. Consequently, the environmental situation can significantly impact changes in the trajectory of the sensorimotor-associative axis [53]. It is due, among other things, to impaired mental and physical health [11–13,51,52,54] and increases school problems [14–16,55]. However, the links between these variables are not clear. Therefore, the present study aims to establish links between depression, anxiety, stress, and the tendency to procrastinate (understood as implemental delay) and school burnout. A literature review allows defining school burnout as an explanatory variable, depression, anxiety, stress as predictors, and procrastination as a mediating variable (mediator). The analyses included four factors of procrastination (implemental delay), and five factors of school burnout. The following hypothesis was formulated: Procrastination (implemental delay) is a significant mediator of the relationship between depression, anxiety, stress and school burnout.

## Material and method

The study aimed to establish links between depression, anxiety, stress and the tendency to procrastinate (implemental delay) and school burnout, hence high school and technical school students were examined.

### Subjects

The study was conducted in April-May 2023. Participation was voluntary and confidential, data processing was anonymous, and only students whose parents gave written consent took part in the study. Since the research was conducted in schools, during parenting lessons,

teachers were asked to support the implementation of the study. They were trained in the distribution and collection of questionnaires. All questionnaires, once completed by the students, were secured and stored in accordance with the recommendations of the Data Protection Law. The Bioethics Committee of the WSEI University approved the study (number 01/03/2023, dated 17/03/2024).

## Procedure

The survey was conducted face-to-face during educational lessons. Completing and collecting the questionnaires and data sheets took about 1 lesson hour (45 minutes). 400 students participated in the survey. After removing records filled out incompletely and incorrectly (using the *listwise* removal method), data from 344 students were analyzed, including 198 girls (57.6%) and 146 boys (42.4%). It is withing the range of the allowed 5% maximum error of the required sample size for the survey (384 subjects). The study comprised 50% high school and 50% technical school students (172 in each group). The group of high school students consisted of 114 (66.3%) girls and 58 (33.7%) boys, while the group of technical school students included 84 (48,8%) girls and 88 (51.2%) boys. The age of the respondents ranged from 14–20 (16.69 ± 1.74), of which 214 (62.2%) were of 14–16 age, while 130 (37.8%) were of 17–20 age. The largest number of students came from large cities—118 (34.3%) and from rural areas—114 (33.1%), medium-sized cities were represented by 78 (22.7%), while small cities were represented by 34 respondents (9.9%).

## Tools

Three standardized questionnaires were used:

**Depression, anxiety and stress scale.** We used version 42 (DASS-42 [56]). The questionnaire contains 42 statements, organized into 3 factors: depression, anxiety and stress. The participants are required to respond to each statement by selecting the degree to which its content refers to their state during the week preceding the measurement. Assessment is made on a 4-point Likert-type scale (from 0 - "it didn't apply to me at all", to 3 - "it applied to me very much or most of the time").

The α-Cronbach's reliability coefficients for the normalization tests were: for the depression scale .93 (in the study group .94); for the anxiety scale .89 (in the study group .90); for the stress scale .92 (in the study group .92) and for the total score .96 (in the study group .97).

The mean and median values showed that more than half of the surveyed students experienced stress, struggle with depression and anxiety problems. Kolmogorov-Smirnov test revealed skewed distribution for all scale items ($p < .001$). The standard error for skewness is .131, for kurtosis is .262.

**Implemental delay scale.** The scale [28] is composed of 13 items including 3 factors: Onset; Sustain; Timeliness. It is also possible to calculate an overall score. A 5-point Likert scale (from 1 - "definitely no", to 5- "definitely yes") is used. The following α-Cronbach's reliability indices were obtained for the sample: Onset = .88; Sustain = .80; Timeliness = .77; for all scale items = .87.

The mean and median values showed that most of the surveyed students did not report problems related to starting activities and tasks. On the other hand, reduced persistence, which translates into problems with the consistent completion of tasks, was admitted by nearly half of them, but this applied mainly to students in the 14–16 age group. More than half of the students indicated problems with completing intended activities or tasks on time. The value of The Kolmogorov-Smirnov statistic showed skewed distribution ($p < .001$) for the extracted scale factors. The standard error for skewness was .131, and .262 for kurtosis.

**Table 1. Descriptive statistics.**

|  | *M* | *MD* | *SD* | *Z(344)* | *SKE* | *Kurt* |
|---|---|---|---|---|---|---|
| Depression | 15.169 | 14.00 | 11.108 | .091 | .518 | -.715 |
| Anxiety | 14.363 | 13.00 | 9.787 | .095 | .430 | -.813 |
| Stress | 18.520 | 17.00 | 10.284 | .070 | .301 | -.751 |
| Onset | 13.773 | 14.00 | 3.829 | .103 | -.255 | -.615 |
| Sustain | 20.061 | 20.00 | 5.040 | .072 | .105 | -.580 |
| Timeliness | 7.389 | 7.00 | 3.202 | .130 | .649 | -.163 |
| Implemental delay / total score | 41.224 | 41.00 | 9.748 | .054 | .137 | -.340 |
| Burnout from Studying (BFS) | 31.244 | 31.00 | 6.084 | .067 | .188 | -.410 |
| Burnout from Family (BFF) | 11.776 | 12.00 | 3.770 | .079 | .214 | -.586 |
| Incompetence in School (IIS) | 11.372 | 11.00 | 3.131 | .077 | .706 | 5.598 |
| Loss of Interest in School (LIS) | 14.988 | 15.00 | 3.153 | .075 | -.366 | .020 |
| Total score | 69.381 | 69.00 | 11.623 | .040 | .231 | -.302 |

**Secondary School Burnout Scale (SSBS).** The SSBS [57] contains 34 items to which respondents answer on a 4-point Likert scale (from 4—strongly agree to 1—strongly disagree). The tool assesses the global level of school burnout (SSBS) and its dimensions: Burnout from Homework (BFH), Burnout from Teacher Attitudes (BFTA), Need to Rest and Time for Fun (NRTF), Loss of Interest to School (LIS), Burnout from Studying (BFS), Burnout from Family (BFF), and Incompetence in School (IIS). Four subscales were used in the analyses: LIS, BFS, BFF, and IIS, and in addition, an overall score.

The α-Cronbach's reliability coefficients for normalization tests were .38-.77 (in the study group .82) for the BFS scale; .67-.79 (in the study group .81) for BFF; .72-.76 (in the study group .70) for IIS; .74-.84 (in the study group .77) for LIS, and, .76-.89 (in the study group .86) for the total score. Scores of more than half of the participants suggested experiencing school burnout. The majority of the students did not report burnout, the underlying cause of which would be parental pressure. At the same time, they did not report a loss of interest in school. Yet more than half of them reported exhaustion from school activities. The Kolmogorov-Smirnov test revealed skewed distribution both for global and for individual scores (p < .001).

Table 1 summarized descriptive statistics for all analyzed variables.

## Results

A mediation analysis [58] was conducted to verify the hypothesis, which assumes that implemental delay (procrastination) is a significant mediator in the relationship between depression, anxiety, stress, and school burnout. In the first step, a series of simple regression analyses were performed between the variables (see Table 2). No significant result was obtained only for the explanatory variable (*Anxiety*) in relation to the explained variable (*Burnout from Studying*).

In the next step. multivariate regression was performed, introducing one of the explanatory variables (anxiety, depression or stress) as the mediator procrastination (implemental delay). The results are summarized in Table 3 and Fig 1. Interestingly, the value of the adopted mediator (implemental delay) was reduced while the value of the predictor increased significantly. This effect is illustrated in Fig 2, while Table 3 presents the values.

The analyses confirmed that procrastination (implemental delay) and its three dimensions: onset, sustain, timeliness are significant mediators of the relationship between stress and burnout from school, and that implemental and sustain delays are mediators of the relationship between depression and burnout from studying. The results of the Sobel test confirmed complete mediation for the relationship between stress and burnout from studying when the

**Table 2. Depression, anxiety, and stress as predictors of implemental delay and school burnout.**

| | Depression | | | | | Anxiety | | | | | Stress | | | | |
|---|---|---|---|---|---|---|---|---|---|---|---|---|---|---|---|
| | $R^2$ | β | F(1.342) | T | p | $R^2$ | β | F(1.342) | T | p | $R^2$ | β | F(1.342) | t | p |
| ID_O | .149 | .386 | 59.798 | 7.733 | <.001 | .112 | .335 | 43.269 | 6.578 | <.001 | .129 | .359 | 50.473 | 7.104 | <.001 |
| ID_S | .158 | .398 | 64.391 | 8.024 | <.001 | .106 | .326 | 40.675 | 6.378 | <.001 | .142 | .377 | 56.779 | 7.535 | <.001 |
| ID_T | .048 | .219 | 17.230 | 4.151 | <.001 | .036 | .191 | 12.882 | 3.589 | <.001 | .029 | .172 | 10.373 | 3.221 | <.001 |
| ID | .184 | .429 | 77.279 | 8.791 | <.001 | .132 | .363 | 51.845 | 7.200 | <.001 | .154 | .392 | 62.226 | 7.888 | <.001 |
| BFS | .053 | .231 | 19.222 | 4.384 | <.001 | .009 | .096 | 3.200 | 1.789 | .075 | .017 | .131 | 5.970 | 2.443 | .015 |
| BFF | .074 | .273 | 27.483 | 5.242 | <.001 | .106 | .326 | 40.599 | 6.372 | <.001 | .070 | .264 | 25.619 | 5.062 | <.001 |
| IIS | .205 | .452 | 87.985 | 9.380 | <.001 | .147 | .383 | 58.892 | 7.674 | <.001 | .144 | .379 | 57.475 | 7.581 | <.001 |
| LIS | .105 | .325 | 40.302 | 6.348 | <.001 | .043 | .209 | 15.551 | 3.944 | <.001 | .044 | .210 | 15.726 | 3.966 | <.001 |
| Total | .176 | .419 | 72.898 | 8.538 | <.001 | .100 | .316 | 37.911 | 6.157 | <.001 | .098 | .313 | 37.213 | 6.100 | <.001 |

Note: ID_O–Onset; ID_S–Sustain; ID_T—Timeliness; ID–Implemental delay; BFS—Burnout from Studying; BFF—Burnout from Family; IIS—Incompetence in School; LIS—Loss of Interest in School; Total–Total score.

**Table 3. Mediation analysis.**

| Model | non-standardized coefficients | | standardized coefficients | | |
|---|---|---|---|---|---|
| | B | SE | β | t | p |
| Burnout from Studying | | | | | |
| Depression | .008 | .028 | .015 | .298 | .766 |
| Implemental delay | .313 | .032 | .501 | 9.712 | <.001 |
| $R^2$ = .258; F(2.341) = 59.390 | | | | | |
| Depression | .006 | .027 | .011 | .222 | .825 |
| Sustain | .667 | .059 | .552 | 11.260 | <.001 |
| $R^2$ = .310; F(2.341) = 76.536 | | | | | |
| Stress | -.048 | .030 | -.081 | -1.599 | .111 |
| Implemental delay | .337 | .032 | .540 | 10.685 | <.001 |
| $R^2$ = .264; F(2.341) = 61.054 | | | | | |
| Stress | .006 | .032 | .011 | .194 | .846 |
| Onset | .533 | .087 | .336 | 6.152 | <.001 |
| $R^2$ = .115; F(2.341) = 22.229 | | | | | |
| Stress | -0.55 | .029 | -.092 | -1.906 | .057 |
| Sustain | .714 | .058 | .591 | 12.235 | <.001 |
| $R^2$ = .317; F(2.341) = 79.133 | | | | | |
| Stress | .052 | .031 | .088 | 1.670 | .096 |
| Timeliness | .474 | .100 | .250 | 4.728 | <.001 |
| $R^2$ = .078; F(2.341) = 14.349 | | | | | |
| Burnout from Family (BFF) | | | | | |
| Depression | .088 | .018 | .258 | 4.845 | <.001 |
| Timelines | .079 | .063 | .069 | 1.255 | .210 |
| $R^2$ = .079; F(2.341) = 14.552 | | | | | |
| Stress | .092 | .019 | .250 | 4.735 | <.001 |
| Timeliness | .095 | .062 | .080 | 1.532 | .129 |
| $R^2$ = .076; F(2.341) = 14.018 | | | | | |

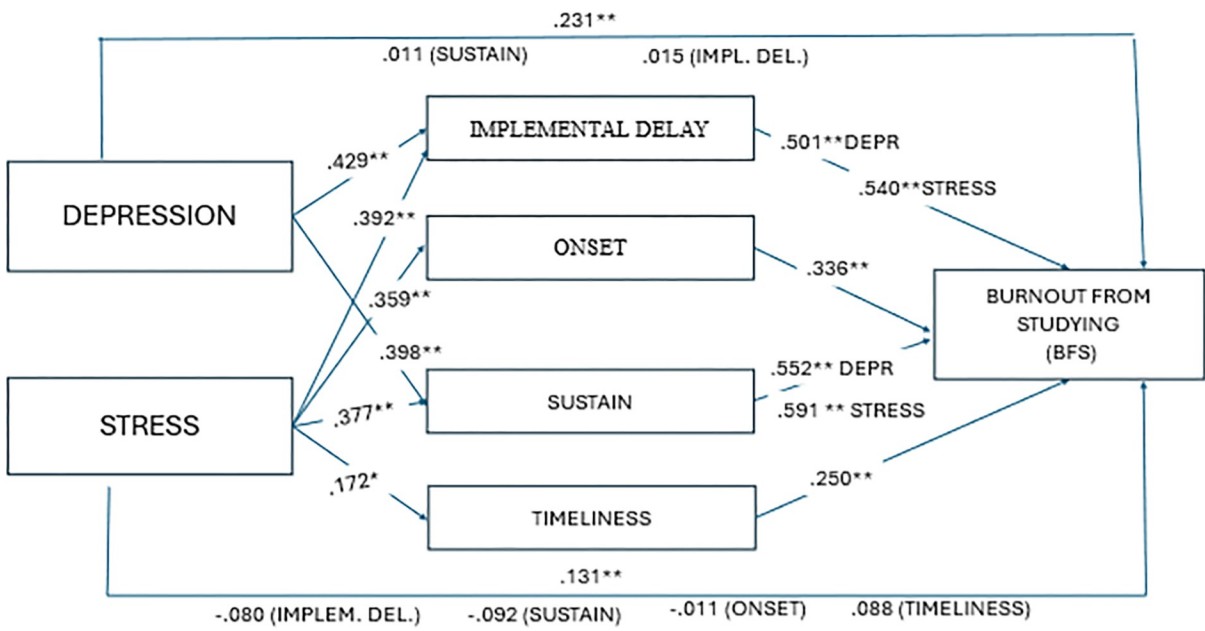

**Fig 1. Mediation results confirmed by Sobel test.**

mediators were implemental delay ($Z = 6.346$, $p < .001$), onset ($Z = 4.650$, $p < .001$), sustain ($Z = 6.416$, $p < .001$) and timeliness ($Z = 2.662$, $p = .008$). In the relationship between depression and burnout from studying, a significant mediating contribution of implemental delay ($Z = 3.155$, $p = .002$) and sustain ($Z = 2.663$, $p = .008$) was confirmed. Unexpectedly, it turned out that in the case of burnout, resulting from parental pressure, the value of the mediator (implemental delay) significantly weakened, while the value of the predictor increased. Sobel's test confirmed the effect for depression ($Z = 3.152$, $p = .002$) and stress ($Z = 2.663$, $p = .008$). It should also be noted that the vast majority of remaining associations between predictors and

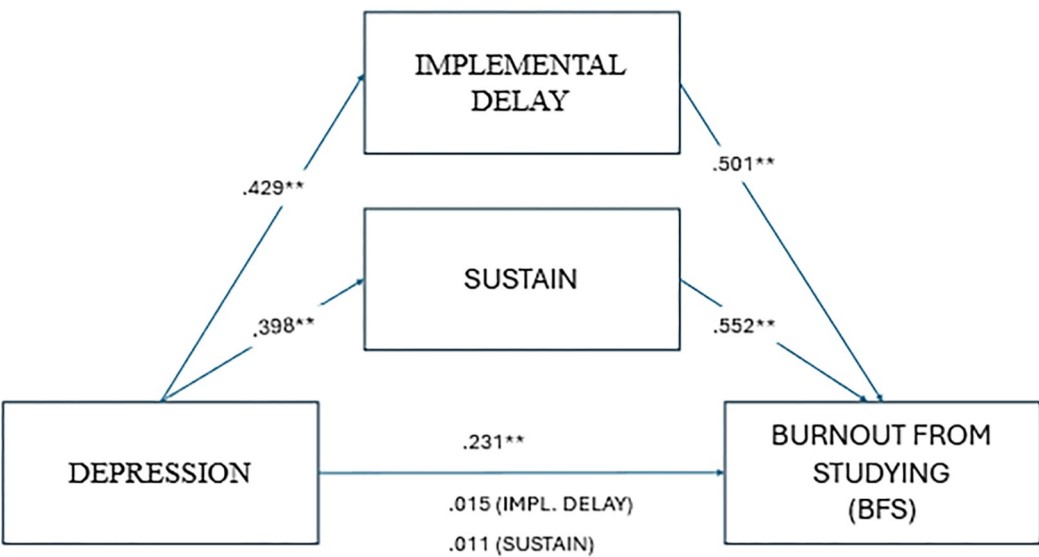

**Fig 2. Depression and stress as mediators of the relationship between timeliness and family situation burnout.**

the mediator (in all its aspects) and school burnout (in all its dimensions) showed a partial mediation effect (both values, predictor and mediator remained significant). Only anxiety was found to be a nonsignificant predictor of burnout from studying ($\beta$ = .096, $p$ = .075).

## Discussion

In summary, the study sought to establish links between depression, anxiety, stress, procrastination behavior (understood as implemental delay) and school burnout. We hypothesized that procrastination (implemental delay) is an essential mediator of the relationship between depression, anxiety, stress and school burnout. Our research shows that stress and depression intensify procrastination tendencies, which then amplify the negative effects associated with school burnout. In addition, three main conclusions emerged from our research. First, we found that procrastination contributes more to school burnout stress or depression. Second, burnout rooted in parental pressure is generated more by depression or stress than by procrastination behavior. Third, depression, anxiety, stress and procrastination are strong predictors of school burnout. As an exception and contrary to our expectations, we found that anxiety did not sufficiently predict burnout from studying.

After the COVID-19 pandemic, there has been an exponential increase in the number of teenagers struggling with mental health problems in Poland. It cannot be ruled out that this effect was reinforced by the events of February 2022, when the war in Ukraine began, and a wave of public concern grew in Poland due to the influx of a wave of emigrants, disturbing and even tragic news reported by the media. Adolescence is a time of exceptional sensitivity and susceptibility to change (both good and bad), which is due, inter alia, to the structural and functional transformation of the maturing brain. An unfavorable environmental situation during adolescence can significantly impact changes in the trajectory of the sensorimotor-associative axis [54], resulting in the disorganization of cognitive processes, which are intensely involved in students' educational outcomes. Many researchers agree that depression, anxiety, stress [50] and procrastination [51] lead to negative consequences. They are associated with the deterioration of mental and physical health [52,59,60] and, in the case of students, compound problems in school learning [56].

However, the links between these variables are not clear. For example, a series of longitudinal studies by Jochmann et al. [10] confirmed the detrimental effects of procrastination on mental health. Yet, they found that procrastination did not lead to perceived stress or depression and anxiety symptoms over time. In contrast, other researchers [9,61] reported procrastination positively to predict academic stress and anxiety. Our study suggests the possibility that it is stress and depression that intensify procrastination tendencies, which amplify the adverse effects associated with school burnout. The strong impact of procrastination on attitudes toward schooling and school as an institution is also argued by Demeter, et al. [62]. The research review we performed shows that many reports refer to the analysis of the impact of school burnout on academic performance [63], stress [64], or depression [65]. The link between procrastination and school burnout is pointed out [66], or the generally advocated need to distinguish between severe and mild types of procrastination and differentiate their impact on students' school careers [6]. Researchers are also looking for the mediating effect of procrastination in the relation to various variables, for example, physical activity and depression [16]. In contrast, information on the mediating contribution of procrastination to the relationship between depression, stress, anxiety, and procrastination and school burnout is not common. Such reports are lacking; if there are, they often relate to teachers' professional activities [67]. Reports on academic burnout in adolescents focus on a single variable, such as stress [68] or study university students [69,70].

The results presented in this report provide an impetus for further research, including those of a clinical nature. It is of particular significance at a time of growing mental health problems in adolescents, especially since prostration is considered one of the causes of these disorders [14–16,59]. Also, many researchers attribute the origins of procrastination to behavioral factors. often linked to reduced control of self-regulation [36,66,70] and pay less attention to the analysis of emotional factors. While the contribution of cognitive-motivational factors may explain procrastination [33,63,70], the negative emotions accompanying procrastination amplify negative attitudes toward schooling and increase the burnout effect leading to low academic achievement [14–16,59]. Therefore, paying more attention to emotional factors both in research work and educational practice might contribute to improving the education system and reducing the incidence of school burnout.

## Author Contributions

**Conceptualization:** Katarzyna Markiewicz, Bożydar L. J. Kaczmarek.

**Data curation:** Katarzyna Markiewicz, Bożydar L. J. Kaczmarek.

**Formal analysis:** Katarzyna Markiewicz, Bożydar L. J. Kaczmarek.

**Funding acquisition:** Katarzyna Markiewicz, Bożydar L. J. Kaczmarek.

**Investigation:** Katarzyna Markiewicz, Bożydar L. J. Kaczmarek.

**Methodology:** Katarzyna Markiewicz, Bożydar L. J. Kaczmarek.

**Project administration:** Katarzyna Markiewicz, Bożydar L. J. Kaczmarek.

**Resources:** Katarzyna Markiewicz, Bożydar L. J. Kaczmarek.

**Software:** Katarzyna Markiewicz, Bożydar L. J. Kaczmarek.

**Supervision:** Katarzyna Markiewicz, Bożydar L. J. Kaczmarek.

**Validation:** Katarzyna Markiewicz, Bożydar L. J. Kaczmarek.

**Visualization:** Katarzyna Markiewicz, Bożydar L. J. Kaczmarek.

**Writing – original draft:** Katarzyna Markiewicz, Bożydar L. J. Kaczmarek.

**Writing – review & editing:** Katarzyna Markiewicz, Bożydar L. J. Kaczmarek.

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
