## [Decision Letter · Decision Letter 0]

15 Oct 2024

PONE-D-24-33145Implemental delay as a mediator of relationship between depression, anxiety, stress and school burnoutPLOS ONE

Dear Dr. Markiewicz,

Thank you for submitting your manuscript to PLOS ONE. After careful consideration, we feel that it has merit but does not fully meet PLOS ONE’s publication criteria as it currently stands. Therefore, we invite you to submit a revised version of the manuscript that addresses the points raised during the review process.

We look forward to receiving your revised manuscript.

Kind regards,

Nicholas Aderinto Oluwaseyi

Academic Editor

PLOS ONE

Additional Editor Comments (if provided):

Reviewers' comments:

Reviewer's Responses to Questions

**Comments to the Author**

1. Is the manuscript technically sound, and do the data support the conclusions?

Reviewer #1: Yes

2. Has the statistical analysis been performed appropriately and rigorously? 

Reviewer #1: Yes

3. Have the authors made all data underlying the findings in their manuscript fully available?

Reviewer #1: Yes

4. Is the manuscript presented in an intelligible fashion and written in standard English?

Reviewer #1: Yes

5. Review Comments to the Author

Reviewer #1: Dear authors,

I consider that this topic is totally relevant in our current research. This study analyzes the role of procrastination in the relationship of depression, anxiety and stress with school burnout in Polish adolescents. The method is appropriate, the results are well described, and data are correctly interpreted. However, this reviewer consider that authors should improve some aspects of the manuscript:

1.Abstract: The percentage of female or male participants should be added in the sample description.

2.The previous literature on the analyzed constructs has been reviewed, but before the Material and method section the need to examine the relationship between them in adolescence should be justified. In addition, the aim and hypothesis that have been written at the beginning of the Material and method section should also appear before this section, that is, at the end of the theoretical review. The hypothesis should include citations of the studies that support it.

3.Discussion: This reviewer suggests referring to the study's hypothesis before discussing the results and deepening the practical implications of the results of this research to enrich final considerations.

Yours faithtully,

The reviewer

6. PLOS authors have the option to publish the peer review history of their article (what does this mean?). If published, this will include your full peer review and any attached files.

Reviewer #1: No

---

## [Author Response · Author response to Decision Letter 0]

21 Nov 2024

First of all, we would like to thank you and the reviewer for carefully reviewing our paper and for your remarks that made it possible to improve it and make it more readable. 

We have also taken into account the reviewer's suggestions and introduced the following amendments, which we marked in blue:

1. We have prepared the paper in accordance with PLOS ONE's style requirements.

2. We have included the data availability address.

3. Both the title on the online submission form (via Edit Submission) and the title in the manuscript are now identical.

4. The percentage of female participants is added in the sample description.

5. We have justified the need to examine the relationship between depression, anxiety, stress and school burnout in adolescence. 

6. Also, the aim and hypothesis have been moved before the Material and method section at the end of the theoretical review. 

7. Citations of the studies that support the hypothesis are included. 

8. We have referred to the study's hypothesis before discussing the results

9. We have discussed the practical implications in more detail.

---

## [Editor Report · Decision Letter 1]

6 Dec 2024

Implemental delay as a mediator of the relationship between depression, anxiety, stress and school burnout

PONE-D-24-33145R1

Dear Dr. Markiewicz,

We’re pleased to inform you that your manuscript has been judged scientifically suitable for publication and will be formally accepted for publication once it meets all outstanding technical requirements.

Kind regards,

Nicholas Aderinto Oluwaseyi

Academic Editor

PLOS ONE
---

## [Editor Report · Acceptance letter]

18 Dec 2024

PONE-D-24-33145R1 

PLOS ONE

Dear Dr. Markiewicz, 

I'm pleased to inform you that your manuscript has been deemed suitable for publication in PLOS ONE. Congratulations! Your manuscript is now being handed over to our production team.

Kind regards, 

on behalf of

Dr. Nicholas Aderinto Oluwaseyi 

Academic Editor

PLOS ONE